# Extending Ag Nanoparticles as Colorimetric Sensor to Industrial Zinc Electrolyte for Cobalt Ion Detection

**DOI:** 10.3390/molecules28020592

**Published:** 2023-01-06

**Authors:** Ni Xiao, Wei Weng, Ding Tang, Wen Tan, Liye Zhang, Zheyuan Deng, Xiaopeng Chi, Jiangang Ku, Shuiping Zhong

**Affiliations:** 1School of Materials Science and Engineering, Fuzhou University, Fuzhou 350108, China; 2Zijin School of Geology and Mining, Fuzhou University, Fuzhou 350108, China; 3Fujian Key Laboratory of Green Extraction and High Value Utilization of New Energy Metals, Fuzhou University, Fuzhou 350108, China; 4Zijin Mining Group Co., Ltd., Shanghang, Longyan 364200, China

**Keywords:** colorimetric detection, silver nanoparticles, polyvinylpyrrolidone, trace Co^2+^ determination, industrial high-concentration zinc solution

## Abstract

The direct and rapid determination of trace cobalt ion (Co^2+^) in the electrolyte of zinc smelting plants is urgently needed but is impeded by the severe interference of extremely high-concentration zinc ions in the solution. Herein, colorimetric detection of Co^2+^ by the polyvinylpyrrolidone functionalized silver nanoparticles (PVP-AgNPs) is realized in solutions with the Zn/Co ratio being high, up to (0.8–5) × 10^4^, which is located within the ratio range in industrial solution. The high concentration of Zn^2+^ induces a strong attenuation of Co^2+^-related signals in ultraviolet-visible (UV-vis) extinction spectra; nevertheless, a good linear range for detecting 1–6 mg/L Co^2+^ in 50 g/L Zn^2+^ solution is still acquired. The strong anti-interference toward other metal ions and the mechanism understanding for trace Co^2+^ detection in such a high-concentration Zn^2+^ solution are also revealed by systematic analysis techniques. The results extend the AgNPs as colorimetric sensors to industrial solutions, providing a new strategy for detecting trace-metal ions in industrial plants.

## 1. Introduction

Colorimetry is a commonly used strategy for the detection of metal ions; however, it is confined to applications in relatively narrow fields [1,2]. The colorimetric method possesses many merits including convenience, rapid response, and low detection limit [3,4]. For example, the silver nanoparticles (AgNPs) suspension, which is a widely used agent for colorimetric detection, can be processed into portable test paper for the qualitative visual inspection or the quantitative spectrophotometry detection of metal ions within a very short timeframe [5,6]. Despite the above superiorities, the reported detection fields for both AgNPs suspension and other colorimetric agents are mainly focused on solutions with a very low total concentration of metal ions, i.e., drinking water, pesticide residue analysis, and sera diagnosis [7,8,9,10]. The above samples contain a relatively low concentration of metal ions, though serum as a special case reached ~4 g/L [11]. For high-concentration solutions, the signals of trace ions can be easily blocked by the dominant metal ions during detection [12]. Therefore, extending the colorimetric method to the detection of trace ions in high-concentration solutions is very challenging, yet is urgently needed.

The zinc smelting plant is one of those areas that is urgently needed for the detection of trace-metal ions in high-concentration solutions. For example, the Zn^2+^ concentration in the electrolyte for industrial electrodeposition of metal zinc is high, up to 50 g/L [13], however, the Co^2+^ must be strictly controlled to be lower than 1.0 mg/L [14]. Fluctuation of Co^2+^ concentration in the high-concentration Zn^2+^ solution will largely decrease the electro-deposition efficiency of metal zinc, even leading to “plate-burning” during the electrodeposition process [15,16]. 

Precise and rapid monitoring of the trace Co^2+^ in the high-concentration Zn^2+^ solution is very important in a zinc smelting plant, but there are great challenges in its implementation. Industrially, detection of Co^2+^ in the Zn^2+^-containing electrolyte is carried out using electrothermal atomic absorption spectrometry with expensive instruments, or spectrophotography which involves cumbersome operation [17,18]. Additionally, the on-site detection and real-time monitoring of Co^2+^ is difficult to conduct because the acid and the noisy, mechanical vibrational environment on site do not allow for the stable functioning of these precise instruments [19,20]. In fact, the tedious analysis procedures, including the sample preparation, transfer of sample from the on-site spot to the analysis-testing center, and sample pretreatment for analysis, have greatly hindered the precise control and swift emergency response of the electrolysis process in the industry.

In recent years, facile and high-efficiency detection of Co^2+^ in zinc electrolyte is explored by the polarography method, which uses a dropping mercury electrode to increase the sensitivity and selectivity of trace Co^2+^ [16,21,22,23]. However, the following challenges restrict its widespread application: (1) the highly toxic property of mercury. The usage of mercury poses a huge threat to both the environment and human health [24]. (2) The detection range of Co^2+^ concentration is not suitable for zinc electrolyte. Zhu Hong-qiu et al. [15] determined the linear range for Co^2+^ detection to be 5.89 × 10^−6^ mg/L–1.89 × 10^−2^ mg/L, showing a huge deviation with the actual concentration of 0.3–2 mg/L. Exploring environmentally friendly and on-site applicable detection methods for Co^2+^ in zinc electrolyte is still of prime importance.

Herein, using the polyvinylpyrrolidone-functionalized AgNPs (PVP-AgNPs) as the colorimetric agent, the detection of trace Co^2+^ (1–6 mg/L) in 50 g/L Zn^2+^ solution is performed. The AgNPs with a strong surface plasmon resonance (SPR) effect were chosen to increase the sensibility for detecting trace Co^2+^ [25,26,27,28]. The anchored PVP was designed to act as a capping agent for linking the target analytes as well as stabilizer for the AgNPs [29]. The mechanism analysis revealed that the anchored PVP plays an important role in grabbing the trace Co^2+^ in the high-concentration solution, therefore, partially counteracting the interference of high-concentration Zn^2+^ in the solution. The results of this paper can provide new insights for the detection of trace-metal ions in high-concentration Zn^2+^ solution.

## 2. Results and Discussion

### 2.1. Interference of Zn^2+^ for Sensing Co^2+^

The PVP-AgNPs present a strong sensing ability toward trace Co^2+^ in Zn^2+^-free solutions. The incubation time between PVP-AgNPs and Co^2+^ was chosen according to the color change of the solution. After mixing different concentrations of Co^2+^ with PVP-AgNPs, a rapid color response occurred, and the color basically did not change after 2 min, so 2 min was selected as the incubation time. As shown in Figure 1a, the color of the solution gradually changes from light yellow to light purple when the added Co^2+^ increases from 0 mg/L to 2 mg/L, implying the outstanding sensing ability of PVP-AgNPs for Co^2+^ in the absence of Zn^2+^. Such a conclusion is also revealed by the UV-vis spectrum in Figure 1b. For single PVP-AgNPs, a strong absorption peak at around 393 nm is observed (Figure 1b), implying the highly dispersed state of the particles [30,31]. After adding Co^2+^, the peak intensity at 393 nm progressively decays with the increase of Co^2+^ concentration, accompanied by the intensity enhancement of a newly generated peak at 533 nm (Figure 1b), which denotes the distance decrease between particles in the solution [32]. The above phenomenon means that adding Co^2+^ induces the aggregation of PVP-AgNPs in the solution, implying that the trace Co^2+^ is grabbed by the PVP chains on the surface of AgNPs [29]. Such results are further quantitatively characterized by the peak intensity ratio, as shown in Figure 1c. A good linear relationship was found between A_533_/A_393_ (intensity ratio for peak at 533 nm and peak at 393 nm) and ln[Co^2+^] (natural logarithm function of Co^2+^ concentration), meaning that the trace Co^2+^ in the Zn^2+^-free solution can be quantitatively determined by the PVP-AgNPs [26,31].

Discouragingly, addition of Zn^2+^ leads to a shocking change in both the color and the UV-vis signals of the solution, meaning a devastating influence on the PVP-AgNPs in terms of sensing the trace Co^2+^. As shown in Figure 2a, addition of Co^2+^ into the PVP-AgNPs suspension turns the color from light yellow to light purple. However, co-addition of Co^2+^ and Zn^2+^ results in the disappearance of the purple color, being replaced with the orange (Co_2_Zn_1K_, namely the solution containing 2 mg/L Co^2+^ and 1 g/L Zn^2+^) or dark yellow (Co_2_Zn_50K_, namely the solution containing 2 mg/L Co^2+^ and 50 g/L Zn^2+^) images. The results indicate that the existence of Zn^2+^ in the solution can interfere with the sensing of trace Co^2+^. This deduction is also demonstrated in the UV-vis spectra. As show in Figure 2b, compared with single Co^2+^, co-existence of Co^2+^ and Zn^2+^ (Co_2_Zn_1K_ and Co_2_Zn_50K_) leads to a dramatic change in the absorption spectrum shape, including peak intensity and position, which implies that Zn^2+^ in the solution influences the aggregation degree of particles to a large extent. Compared with single Zn^2+^ (Zn_1K_ and Zn_50K_), Co_2_Zn_1K_ and Co_2_Zn_50K_ also showed a discernable change in absorption spectrum shape. This means that when the concentration of Zn^2+^ is significantly higher than that of Co^2+^ (the concentration ratio of Zn^2+^ to Co^2+^ is 500:1 and 25,000:1, respectively), the addition of Co^2+^ can still affect the aggregation state of particles.

Therefore, the PVP-AgNPs present a strong ability in colorimetric sensing of the trace Co^2+^ in the Zn^2+^-free solution. Although the existence of high-concentration Zn^2+^ can severely interfere with the detection signals of Co^2+^, the trace amount of Co^2+^ can still affect the absorption spectrum shape and the aggregation state of particles. 

### 2.2. Detection of Trace Co^2+^ in High-Concentration of Zn^2+^

The high-concentration Zn^2+^ induces a strong change in the signal of trace Co^2+^; nevertheless, a good linear relationship is still found between the peak intensity and the Co^2+^ concentration. As shown in Figure 3a, even in the solution with a Zn^2+^ concentration as high as 50 g/L (the concentration of industrial zinc electrolyte), an obvious shoulder peak in the trace Co^2+^ is still observed at about 533 nm, with an intensity that rises linearly with the increase of Co^2+^ concentration (Inset in Figure 3a). The results imply that the interaction between the superficial PVP chains and the trace Co^2+^ still exists, although the majority of active sites on the surface of PVP-AgNPs for sensing activity are consumed by the dominant Zn^2+^ in the solution.

The absorption intensity ratio between the newly appeared peak (533 nm) caused by analytes and the initial PVP-AgNPs peak (393 nm) is commonly used for the quantitative determination of trace ions using the colorimetric method [26,33]. As shown in Figure 3b, a good linear relationship between A_533_/A_393_ and Co^2+^ concentration in the range of 1–6 mg/L is found (y = 0.0162x + 0.1808, R^2^ = 0.9936), which means that the concentration of trace Co^2+^ in the 50 g/L Zn^2+^ solution can be precisely determined. As has been documented, the maximum tolerable level of cobalt is 0.3–2 mg/L for typical zinc smelting plants across China [34], and 0.05–1 mg/L for zinc smelters in other countries [35]. Therefore, the detection range of 1–6 mg/L enables the fabricated PVP-AgNPs to monitor the Co^2+^ concentration during normal operation or alarming for exceeding limits of Co^2+^ concentration in zinc smelting plants. 

Both the fluctuation of Zn^2+^ concentration and the typical trace impurities in the industrial zinc electrolyte have negligible influence on the detection of Co^2+^. For the industrial zinc electrolyte, the concentration of Zn^2+^ is commonly fixed at 50 g/L to enable a stable electrodeposition process. However, fluctuations in the industrial process is inevitable, therefore, the concentration of Zn^2+^ in real zinc electrolyte usually varies between 48–52 g/L. As shown in Figure 4a, variation of the Zn^2+^ concentration between 48–52 g/L leads to fluctuation of the Co^2+^ signal within 10%. Such a signal deviation ratio induced by zinc concentration fluctuation can still guide the process adjustment for zinc electrodeposition in the industry due to the following reasons: (1) Fluctuation in the zinc concentration is the result of accumulation over months, rather than an instant case, which means the influence of zinc fluctuation is very small within the testing periods. In this regard, Gong et al. have also pointed out that in the process of zinc electrolysis, the transition process of zinc concentration in the electrolyte is very slow [36]. (2) Once the Co^2+^ in the solution induces abnormal phenomenon for production, the Co^2+^ concentration commonly increases in multiples, greatly exceeding the signal deviation ratio of 10%. On the one hand, analytical detection methods usually allow a deviation of less than 10% [37]. On the other hand, under normal production conditions, the concentration of Co^2+^ in zinc sulfate electrolyte should not exceed 1 mg/L, but once the Co^2+^ in the solution induces abnormal phenomenon, the Co^2+^ concentration may reach 2 mg/L or even higher [34]. In addition to Co^2+^, other trace ions such as Cu^2+^, Ni^2+^, Ca^2+^, F^−^, Cd^2+^, Fe^2+^, and Pb^2+^ are also devastating in zinc electrolyte. As shown in Figure 4b, the interferences of these trace impurity ions on the detection of Co^2+^ are negligible. The concentrations of interfering ions are selected based on the upper limit or slightly below the upper limit concentration of possible impurity ion in zinc electrolyte [21,22,34]. Therefore, the PVP-AgNPs can be used for the detection of Co^2+^ in industrial zinc electrolyte, although this is faced with the challenges of both the fluctuation of Zn^2+^ concentration and the co-existence of other metal impurity ions.

High-concentration Zn^2+^ induces a strong change for the signal of trace Co^2+^, nevertheless, the precise detection of trace Co^2+^ can still be realized. More importantly, the concentration fluctuation of Zn^2+^ and other typical trace impurity ions in the zinc electrolyte have negligible influence on the detection of trace Co^2+^.

### 2.3. Mechanism Understanding for Co^2+^ Detection in High-Concentration Zn^2+^ Solution

Both the color change and the peak variation in UV-vis spectra of AgNPs suspension can be associated with the aggregation or dispersion states of particles and the characteristics of the surrounding medium [38,39]. According to the available evidence, the aggregation size is an important factor affecting the peak wavelength and intensity of UV-vis spectra. However, there is no evidence for the PVP structure changes caused by the effects of the surrounding dielectric effect, which is beyond the scope of this study. Nevertheless, observing the behavior of PVP-AgNPs in the solution can facilitate understanding of the colorimetric sensing mechanism for Co^2+^.

The TEM images reveal that PVP-AgNPs have an average diameter of 7.8 ± 1.6 nm and in different solutions present entirely different behaviors. As shown in Figure 5a,e, the PVP-AgNPs in high-purity water are highly dispersive, presenting a single characteristic peak at ~393 nm with strong intensity (Figure 1b). After introducing 2 mg/L Co^2+^, obvious aggregation of the particles is observed (Figure 5b,f), which form loose clumps with large cavities, resulting in the generation of a new peak at ~533 nm in UV-vis spectrum and attenuation of the peak at ~393 nm at the same time (Figure 1b). Startlingly, addition of 50 g/L Zn^2+^ leads to a more compact aggregation of PVP-AgNPs than that of 2 mg/L Co^2+^ (Figure 5c,g), which explains why the high-concentration Zn^2+^ has a strong interference in the detection of trace Co^2+^ by PVP-AgNPs. Compared with the PVP-AgNPs dispersion containing sole 50 g/L Zn^2+^, further addition of 2 mg/L Co^2+^ contributes to further gathering of aggregated PVP-AgNPs, resulting in more compact clusters (Figure 5d,h). Therefore, although existence of high-concentration Zn^2+^ alters the dispersion state of PVP-AgNPs, the further addition of trace Co^2+^ can still trigger the signal changes at ~533 nm (Figure 3a), enabling the efficient detection of trace Co^2+^ in high-concentration Zn^2+^ solution. The evolution of the particle dispersion state in various solutions is illustrated in Figure 5i.

The above phenomenon is also manifested by the elements distribution in the TEM results. As shown in Figure 6a–c, a homogeneous distribution of Co^2+^ on the surface of PVP-AgNPs is observed after addition of 2 mg/L Co^2+^, which implies that Co^2+^ is strongly grabbed by the PVP-AgNPs, thus resulting in the efficient detection of trace Co^2+^ [40]. Similarly, adhesion of Zn^2+^ on the surface of PVP-AgNPs is also observed (Figure 6d–f), indicating that Zn^2+^ can be also trapped by the PVP-AgNPs, which explains the strong interference of high-concentration Zn^2+^ for sensing trace Co^2+^ by PVP-AgNPs [41]. It should be mentioned that, for co-existed 50 g/L Zn^2+^ and 2 mg/L Co^2+^, the dominantly high-concentration Zn^2+^ (25,000 times that of Co^2+^) cannot repel the bonding of trace Co^2+^ with the PVP-AgNPs, as revealed by the element distributions in Figure 6g–j [42]. Therefore, signals can be still found in UV-vis spectra for trace Co^2+^ in high-concentration Zn^2+^ solutions.

The above results imply that the sensing ability of PVP-AgNPs is highly relevant with its surface interactions toward metal ions, which is also demonstrated in the zeta potential results and DLS analysis. As shown in Figure 7a, the PVP-AgNPs dispersion presents a large negative value of zeta potential (−9.91 mV), which increases to −0.42 mV after adding 2 mg/L Co^2+^, again manifesting the adsorption of positive-charged Co^2+^ on the negative-charged surface of PVP-AgNPs. Additionally, adsorption of Zn^2+^ takes place after the addition of 50 g/L Zn^2+^, as revealed by the similar increase of zeta potential from −9.91 mV to 0.19 mV. Importantly, introducing 2 mg/L Co^2+^ into the 50 g/L Zn^2+^ solution induces the further increase of zeta potential from 0.19 mV to 0.3 mV, demonstrating that the adsorption of Co^2+^ in the high-concentration Zn^2+^ solution (50 g/L) can still proceed smoothly [32].

The adsorption of metal ions on the surfaces of PVP-AgNPs propels the aggregation of the nanoparticles. As shown in Figure 7b, the particle size of initial PVP-AgNPs is only 22.3 nm. The addition of 2 mg/L Co^2+^ triggers the aggregation of nanoparticles, largely increasing the size to 217.6 nm. Compared with the 2 mg/L Co^2+^, addition of 50 g/L results in a large size, 436.9 nm, due to a more compact aggregation state (Figure 6b and Figure 7b). Co-addition of 2 mg/L Co^2+^ and 50 g/L Zn^2+^ further increases the size to 460.4 nm. The size evolution trend is consistent with that of the zeta potential, proving that aggregation of the nanoparticles is highly dependent on the surface adsorption behaviors. The value discrepancy of particle size between the DLS results and TEM images is due to different measuring theories [43].

The surface interactions for enabling the sensing of trace Co^2+^ in high-concentration Zn^2+^ solution is again manifested by the XPS results. As shown in Figure 8a, the initial PVP-AgNPs solution only had a symmetric single peak at 399.79 eV, corresponding to the C–N bond of the coating agent PVP [44,45]. Addition of Co^2+^ induces the generation of a new Co–N peak at 400.24 eV (Figure 8a) [46], with the bonded nitrogen being increased from 6.7% for 2 mg/L Co^2+^ to 13.8% for 5 mg/L Co^2+^. The results indicate that the added Co^2+^ is bonded with the PVP on the surface of the AgNPs, inducing the aggregation of PVP-AgNPs [29]. Similarly, the added Zn^2+^ also bonded with the superficial PVP on AgNPs, as shown by the generation of a Zn–N bond at 401.43 eV (Figure 8b) [47]. The concentration of added Zn^2+^ (50 g/L) is much higher than that of Co^2+^ (2 mg/L), which provides more metal ions for bonding with the nitrogen in PVP for the former case, thus resulting in a much higher peak intensity of Zn–N than that of Co–N. Obviously, the competing bonding with nitrogen between Zn^2+^ and Co^2+^ means that the detection of trace Co^2+^ by PVP-AgNPs can be interfered with by the dominant Zn^2+^. Compared with solely adding 50 g/L Zn^2+^, further introducing 5 mg/L Co^2+^ again generates the Co–N peak (Figure 8c), which indicates that the smooth bonding of trace Co^2+^ with PVP-AgNPs in high-concentration Zn^2+^ solution can still proceed. Therefore, the sensing detection of Co^2+^ is still feasible in high-concentration Zn^2+^ solution. It should be mentioned that the bonded nitrogen by Co^2+^ decreases from 13.8% in 5 mg/L Co^2+^ solution (Figure 8a) to 10.1% in 5 mg/L Co^2+^+ 50g/L Zn^2+^ solution (Figure 8c), again substantiating that occupation of the active nitrogen-containing sites in PVP by Zn^2+^ can mitigate the interaction between Co^2+^ and PVP-AgNPs. 

High-concentration Zn^2+^ in solution dominantly bonds with the nitrogen in PVP on the surface of AgNPs, which leads to the severe aggregation of PVP-AgNPs, posing an obvious interference on the colorimetric detection of trace Co^2+^. As reported by some researchers, Co^2+^ has a highly flexible bond length and geometry with a maximum coordination number of six (Figure 9a) [26,29,48], while Zn^2+^ is four-coordinated [49] with a relatively rigid coordination geometry (Figure 9b). Therefore, the nitrogen in PVP-AgNPs presents a stronger bonding ability for Co^2+^ than Zn^2+^, therefore retaining the ability for colorimetric detection of trace Co^2+^ even in solutions with high-concentration Zn^2+^. More specifically, the dominant Zn^2+^ and trace Co^2+^ compete for bonding with the active nitrogen-containing sites in PVP-AgNPs, mitigating the signal response of Co^2+^ for colorimetric detection. Although being highly affected by the high-concentration Zn^2+^, smooth bonding of trace Co^2+^ can still occur because of the high bonding ability of Co^2+^ toward nitrogen in PVP-AgNPs, thus still providing the PVP-AgNPs with the sensing ability for trace Co^2+^.

## 3. Materials and Methods

### 3.1. Materials and Reagents

Silver nitrate (AgNO_3_, ≥99.8%) was purchased from Sinopharm Chemical Reagent Co., Ltd. (Shanghai, China). NaBH_4_ (≥98.0%), ZnSO_4_·7H_2_O (≥99.5%), Ca(NO_3_)_2_·4H_2_O (≥99.0%), and FeSO_4_·7H_2_O (≥99.0%) were acquired from Xilong Scientific Co., Ltd. (Shantou, China). PVP (M_w_ ≈ 40,000 g/mol) was purchased from Sigma-Aldrich (St. Louis, MO, USA). CoSO_4_·7H_2_O (≥99.0%), CuSO_4_·5H_2_O (≥99.0%), NiSO_4_·6H_2_O (≥99.0%), NaF (≥98.0%), CdSO_4_·8/3H_2_O (≥99.0%), and PbCl_2_ (≥99.0%) were obtained from Aladdin Biochemical Technology Co., Ltd. (Shanghai, China). Ultra-pure water was obtained by a Direct-Q 3 UV water purification system (Millipore, Burlington, MA, USA) and used throughout. All chemicals were used as received without further purification. All metal ion stock solutions were prepared in ultra-pure water.

### 3.2. Preparation of PVP-AgNPs

The AgNPs were produced by reducing AgNO_3_ in NaBH_4_ aqueous solution with PVP used as a capping agent. The 1% PVP (solution A) and 1 mmol/L AgNO_3_ (solution B) were prepared by adding accurately weighed solid-state PVP and AgNO_3_ powder into ultra-pure water and stirred moderately with a magnetic stirring apparatus. The 2 mmol/L NaBH_4_ (solution C) was prepared using fresh ice water and for immediate use. In a typical synthesis procedure, solution C was mixed with solution A in an ice-water bath with moderate magnetic stirring (500 rpm for 5 min). Subsequently, solution B was added into the mixture of solution A and C at the rate of 4 mL/min by injection pump. After all of the AgNO_3_ solution was added, the stirring was stopped and the solution was transferred into the refrigerator (4 °C). The color of the solution changed from pale yellow to dark yellow after 2 h.

### 3.3. Sensing Studies

The sensing of Co^2+^ was performed at room temperature. Briefly, 2.4 mL of Co^2+^-containing solution of different concentrations and 0.6 mL PVP-AgNPs dispersion were mixed in a colorimetric tube. After incubating for 2 min, the samples were analyzed using an optical spectrometer (UV2600i, Shimadzu, Kyoto, Japan). In the absorption spectroscopy, the peak intensity at 533 nm (A_533 nm_) and the peak intensity at 393 nm (A_393 nm_) represent the degree of aggregation and dispersion of nanoparticles, respectively. The absorbance ratio of A_533 nm_/A_393 nm_ was used as the parameter for Co^2+^ detection [26,50].

### 3.4. Characterization

The morphologies of PVP-AgNPs in the absence and presence of Co^2+^ or Zn^2+^ were characterized using transmission electron microscopy (TEM, Tecnai G2 F20, FEI, Hillsboro, OR, USA). The elemental mapping images were analyzed using energy dispersive X-ray spectroscopy (EDS) attached to TEM apparatus. The surface chemistries were characterized using X-ray photoelectron spectra (XPS) (ESCALAB 250 Xi, Thermo Fisher Scientific, Waltham, MA, USA), with results being calibrated by C 1s at 284.8 eV. The dynamic light scattering (DLS) measurements and zeta potential were analyzed using the Malvern Zetasizer Nanoseries instrument (ZS90, Malvern, UK). The diameter of the 200 particles was calculated from the TEM images, with the aid of the Nano Measurer 1.2 software (Fudan University, Shanghai, China).

## 4. Conclusions

The possibility of using PVP-AgNPs for the colorimetric sensing of trace Co^2+^ in industrial zinc electrolyte with high Zn^2+^ concentration was systematically investigated. In the absence of Zn^2+^, Co^2+^ alone can induce obvious concentration-dependent color change in PVP-AgNPs, from yellow to purple. The addition of high-concentration Zn^2+^ leads to both the color variation and signal attenuation in UV-vis spectra for Co^2+^. Even so, the UV-vis spectra still promise a good linear relationship between the signal intensity and concentration of Co^2+^ in the high-concentration Zn^2+^ solution, namely 1–6 mg/L Co^2+^ in 50 g/L Zn^2+^ solution. Moreover, the fluctuation of Zn^2+^ concentration and the co-existence of other typical impurity ions show negligible influence on this plausible linear relationship for trace Co^2+^ detection. The morphology observation, surface chemistry, and particle size distribution together show that the dominant high-concentration Zn^2+^ competingly bonds with the active nitrogen-containing sites on PVP-AgNPs, therefore affecting the particle aggregation behavior and color variation as well as UV-vis response for the colorimetric detection of trace Co^2+^. However, the interference of dominant Zn^2+^ in the solution cannot stop the bonding of Co^2+^ with the nitrogen in PVP-AgNPs, which enables the smooth detection of trace Co^2+^ in high-concentration Zn^2+^ solution, although with a much weakened signal response. The results may provide new insights for trace Co^2+^ detection for industrial zinc electrolyte with an extremely high Zn^2+^ solution.

## Figures and Tables

**Figure 1 molecules-28-00592-f001:**
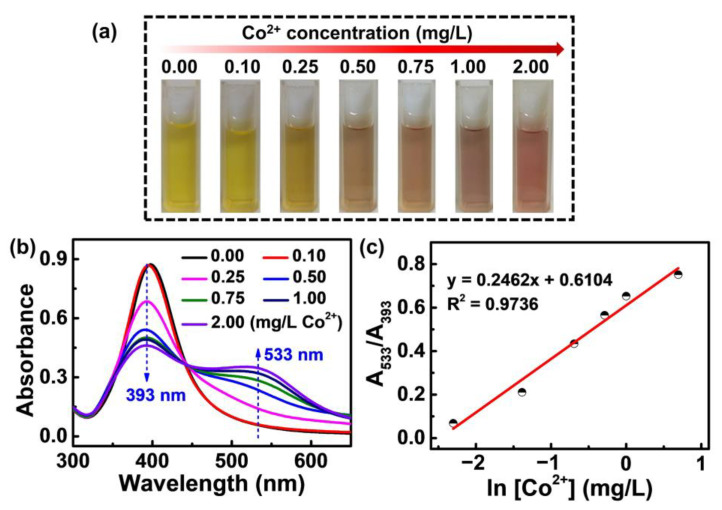
(**a**) Photographic images of PVP-AgNPs sensing solutions with various Co^2+^ concentrations (0.00, 0.10, 0.25, 0.50, 0.75, 1.00, 2.00 mg/L). (**b**) The corresponding absorption spectra of (**a**). (**c**) Plot of A_533_/A_393_ for PVP-AgNPs experienced with different concentrations of Co^2+^.

**Figure 2 molecules-28-00592-f002:**
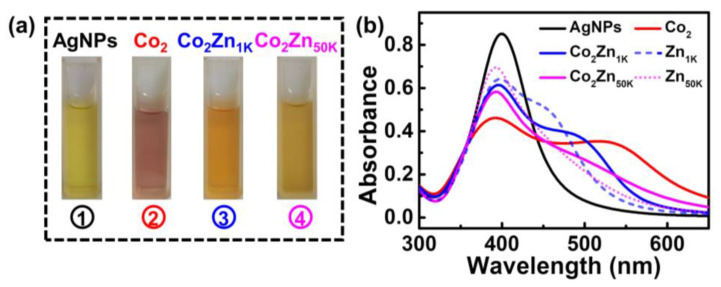
(**a**) Photographic images of PVP-AgNPs sensing solutions: ① PVP-AgNPs; ② PVP-AgNPs + 2 mg/L Co^2+^; ③ PVP-AgNPs + 2 mg/L Co^2+^ + 1 g/L Zn^2+^; ④ PVP-AgNPs + 2 mg/L Co^2+^ + 50 g/L Zn^2+^ (from left to right). (**b**) Solid line (black, red, blue, and pink color): the corresponding absorption spectra of (**a**), short dash (blue): AgNPs + 1 g/L Zn^2+^, short dot (pink): AgNPs + 50 g/L Zn^2+^.

**Figure 3 molecules-28-00592-f003:**
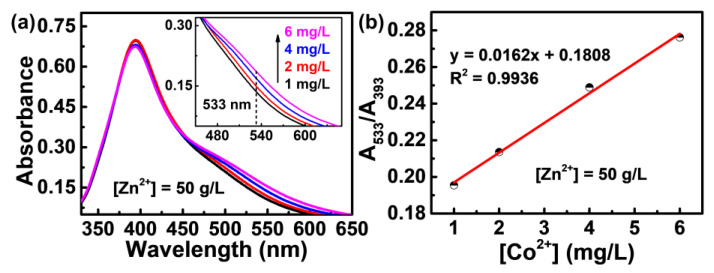
(**a**) UV-vis absorption spectra of PVP-AgNPs mixed with fixed 50 g/L Zn^2+^ and different concentration of Co^2+^ (black line: 1 mg/L, red line: 2 mg/L, blue line: 4 mg/L, pink line: 6 mg/L). The inset shows the enlarged view of 450–650 nm wavelength range. (**b**) The dependence of the A_533_/A_393_ values of PVP-AgNPs on the increasing concentration of Co^2+^.

**Figure 4 molecules-28-00592-f004:**
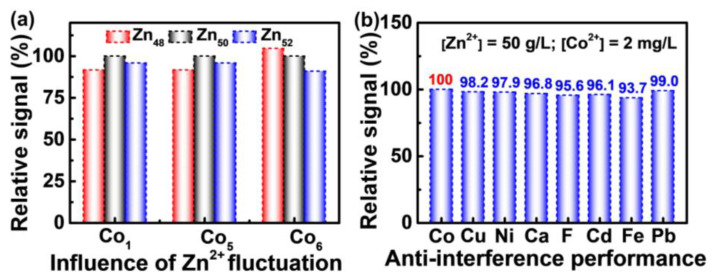
(**a**) Effect of Zn^2+^ concentration fluctuation on Co^2+^ response signal. (**b**) The influence of interfering ions: [Zn^2+^] = 50 g/L, [Co^2+^] = 2 mg/L, [Cu^2+^] = 0.5 mg/L, [Ni^2+^] = 1 mg/L, [Ca^2+^] = 800 mg/L, [F^−^] = 50 mg/L, [Cd^2+^] = 0.1 mg/L, [Fe^2+^] = 2 mg/L, [Pb^2+^] = 0.05 mg/L. The Y-axis “Relative signal (%)” represents the relative values of absorbance ratio between reference and experimental values ((A_533_/A_393_)_ref_/(A_533_/A_393_)_exp_): (**a**) 50 g/L Zn^2+^ was set as reference, and 48 g/L and 52 g/L as the experimental group; (**b**) the mixture of 50 g/L Zn^2+^ and 2 mg/L Co^2+^ was set as reference, a third metal ion introduced into the Zn−Co mixture system was set as the experimental group.

**Figure 5 molecules-28-00592-f005:**
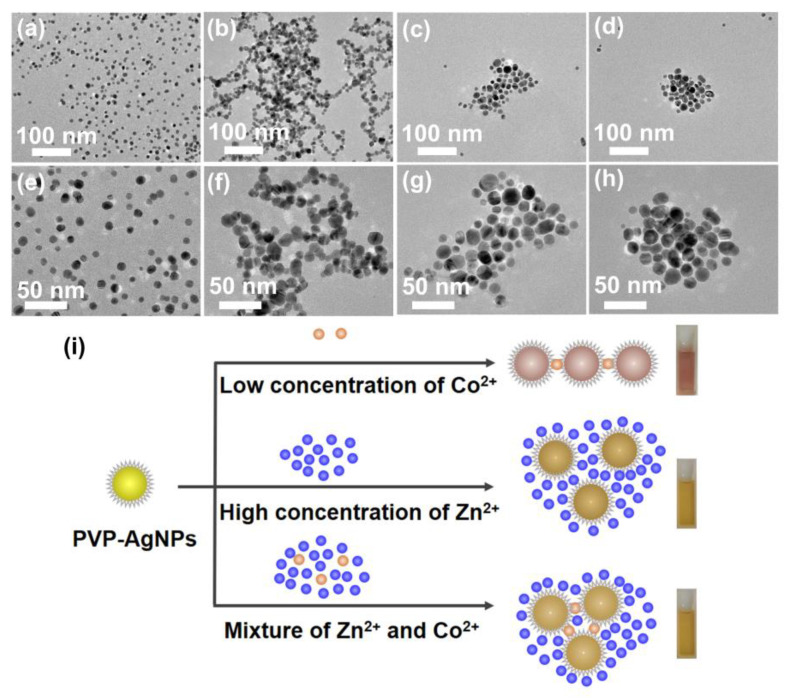
TEM micrograph and size distributions: (**a**,**e**) PVP-AgNPs in pure water free of metal ions; (**b**,**f**) PVP-AgNPs in the presence of 2 mg/L Co^2+^; (**c**,**g**) PVP-AgNPs in the presence of 50 g/L Zn^2+^; (**d**,**h**) PVP-AgNPs in the co-existence of 2 mg/L Co^2+^ and 50 g/L Zn^2+^. (**i**) Illustration of the particle behaviors of PVP-AgNPs in various solutions.

**Figure 6 molecules-28-00592-f006:**
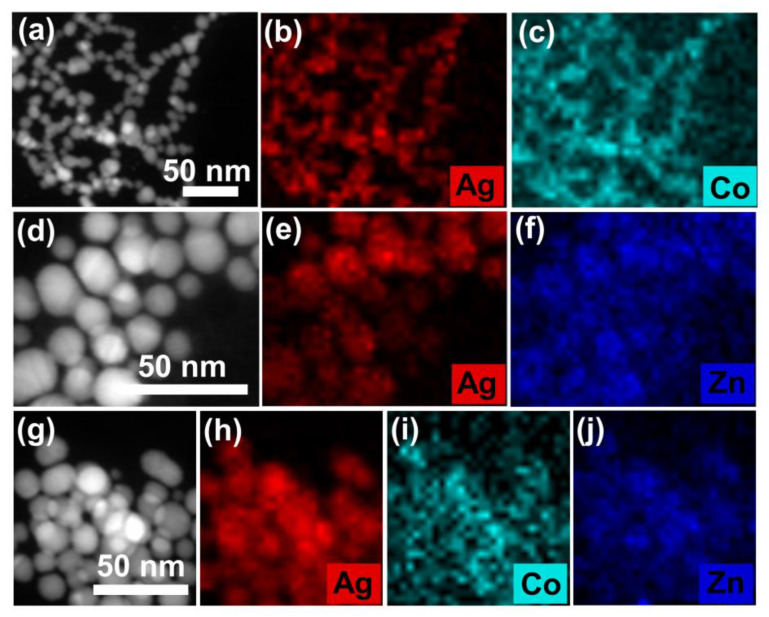
Elemental mappings of PVP-AgNPs in the presence of (**a**–**c**) 2 mg/L Co^2+^, (**d**–**f**) 50 g/L Zn^2+^, (**g**–**j**) mixed solution of 2 mg/L Co^2+^ and 50 g/L Zn^2+^.

**Figure 7 molecules-28-00592-f007:**
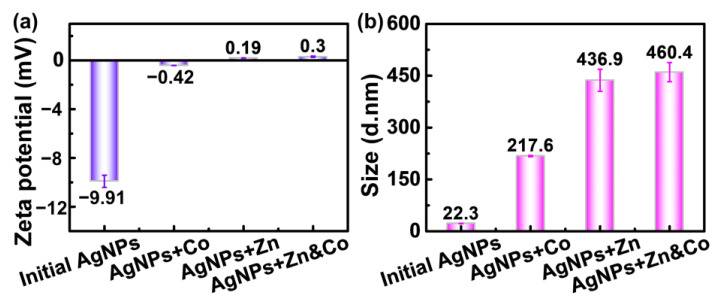
Zeta potential (**a**) and dynamic light scattering (**b**) of PVP-AgNPs before and after the addition of Co^2+^, Zn^2+^, and mixed solution of Co^2+^ and Zn^2+^. [Co^2+^] = 2 mg/L, [Zn^2+^] = 50 g/L.

**Figure 8 molecules-28-00592-f008:**
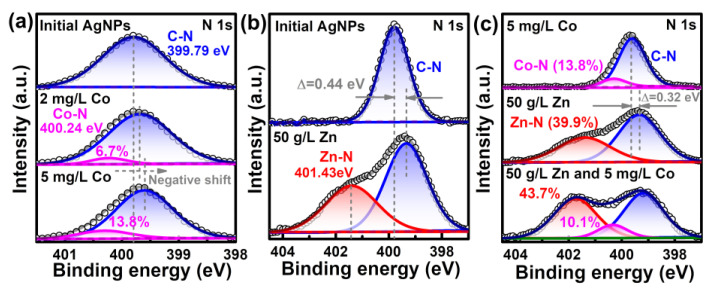
High-resolution XPS spectra comparisons of N 1s: (**a**) PVP-AgNPs dispersions with or without Co^2+^; (**b**) PVP-AgNPs dispersions with or without Zn^2+^; (**c**) PVP-AgNPs dispersions containing Co^2+^, Zn^2+^, or Co^2+^–Zn^2+^ co-existed ions.

**Figure 9 molecules-28-00592-f009:**
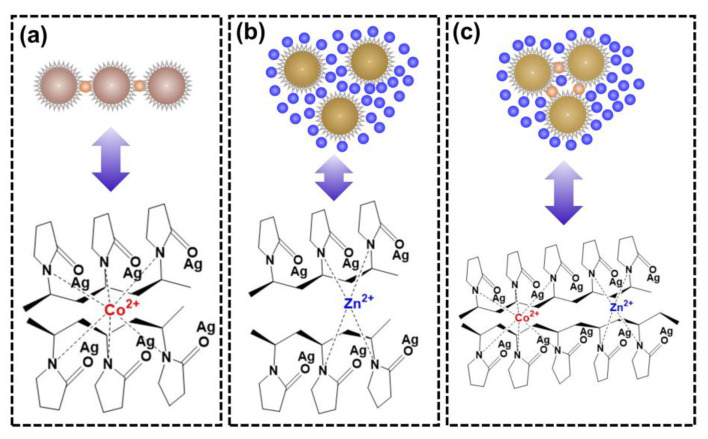
The coordination modes of PVP-AgNPs with metal ions in various solutions: (**a**) Co^2+^; (**b**) Zn^2+^; (**c**) mixture of Co^2+^ and Zn^2+^.

## Data Availability

The data presented in this study are available on reasonable request from the corresponding author.

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
