# Peer review of "Extending Ag Nanoparticles as Colorimetric Sensor to Industrial Zinc Electrolyte for Cobalt Ion Detection"

_molecules, 2023, doi:10.3390/molecules28020592_

Round 1
Reviewer 1 Report
Manuscript - article:
Extending Ag Nanoparticles as Colorimetric Sensor to Indus3 trial Zinc Electrolyte for Cobalt Ion Detection
Autors:
by Xiao et al.
Journal:
Molecules
The revision of the manuscript is required before acceptance.
Comments are below:
1. line 307: "After incubating for 2 min..." - did authors optimized incubation time between PVP-AgNPs and Co2+ ? Please explained.
2. I suggest that the authors characterize PVP-AgNPs material also using the SEM method.
Author Response
Point 1: line 307: "After incubating for 2 min..." - did authors optimized incubation time between PVP-AgNPs and Co2+? Please explained.
Response 1: Thanks for reviewer’s comment. The incubation time between PVP-AgNPs and Co2+ was chosen according to the color change of the solution. After mixing different concentrations of Co2+ with PVP-AgNPs, rapid color response occurred, and the color basically did not change after 2 min, so 2 min was selected as the incubation time. The related comments are added in the revised manuscript (lines 85-88) as follows:
“The incubation time between PVP-AgNPs and Co2+ was chosen according to the color change of the solution. After mixing different concentrations of Co2+ with PVP-AgNPs, rapid color response occurred, and the color basically did not change after 2 min, so 2 min was selected as the incubation time.”
Point 2: I suggest that the authors characterize PVP-AgNPs material also using the SEM method.
Response 2: Thanks for reviewer’s comment. Due to the epidemic COVID-19, our school was closed in advance, so SEM characterization could not be performed during the time of revision.
Reviewer 2 Report
Manuscript Number: molecules-2100819
Molecules Draft
Title: Extending Ag Nanoparticles as Colorimetric Sensor to Industrial Zinc Electrolyte for Cobalt Ion Detection
Recommendation: Minor revisions needed as noted.
Comments:
This manuscript describes the synthesis of PVP-functionalized silver nanoparticles (PVP-Ag). These PVP-Ag nanoparticles are then used for sensing cobalt ion concentration in an industrial zinc electrolyte system. The PVP-Ag nanoparticles coordinate with Co2+ and Zn2+ ions through the PVP which leads to larger aggregates introducing additional peaks in the UV-Vis spectrum around 500 nm. The peak arising in the presence of Co2+ increases with Co2+ concentration in a linear trend, which can be calibrated to sense Co2+ concentration. High concentration of zinc ~50g/L in a typical industrial electrolyte, coordinates strongly to the PVP-Ag and affects the Co2+ peak, which the authors accounted for. The authors characterize the Ag-PVP nanoparticles thoroughly using TEM and DLS and track the sensing mechanism using size changes and zeta potential. XPS studies also corroborate with the zeta potential changes and size changes after coordination induced aggregation. Overall, the authors present a thorough study of PVP-Ag nanoparticles, their utility and mechanism of sensing Co2+ ion concentration even in a highly Zn2+ concentrated media. However, a few details can be addressed better. Below is a list of concerns and suggestions to be considered.
1. General: The authors mention that typical Zn2+ concentration in an industrial electrolyte solution is ~50 g/L, but do not provide typical concentrations for Co2+ found in these solutions. While the Co2+ detection follows a linear trend useful for 1-6 mg/L Co2+ concentration, the authors should discuss how this is relevant to typical industrial electrolyte systems with proper references.
2. General: From the TEM images, there does not seem to be much difference in the aggregates produced in presence of Zn2+ and Zn2++Co2+. XPS studies show that binding is mostly through the PVP ligands, is a different way to the Zn2+. In that case why does Co2+ show a different peak in the UV-Vis spectrum. Is it a totally aggregation size influenced peak wavelength or more of a change in surrounding dielectric effect with changes in PVP structure. A brief explanation on this would enhance the importance of this manuscript.
3. Figure 2b: The UV-Vis spectrum in the presence of just Zn2+ ions is not shown, the authors should include that as a reference here, especially with Zn2+ 1g/L and 50g/L..
4. Figure 4: The Y-axis quantity “Retention Rate %” should be explained a bit more clearly in the text or the figure caption. Perhaps changing the Y-axis title to something like Change in Intensity would convey the meaning better.
5. Figure 4b: The concentrations of all the other interfering ions are widely different with no explanation of why these were selected. The authors should provide an explanation and reference for using these concentrations.
6. Page 4, lines 156-158: “1) Fluctuation of zinc concentration is an accumulating result of months rather than an instant case, which means the influence of zinc fluctuation is very small within the testing periods.” This is a critical point for the relevance of this study, 1-2 references must be provided to substantiate this claim. Similarly for point 2 regarding Co2+“concentration increasing in multiples greatly exceeding the signal deviation ratio of 10%.”
